# GRPO is Secretly a Process Reward Model

Michael Sullivan [1]  Alexander Koller [1]

## Abstract

Process reward models (PRMs) allow for fine-grained credit assignment in reinforcement learning (RL), and seemingly contrast with outcome reward models (ORMs), which assign a single reward to an entire trajectory. However, we provide theoretical proof in this work that the Group Relative Policy Optimization (GRPO) RL algorithm equipped with an ORM is in fact equivalent to a PRM-aware RL objective equipped with a non-trivial, Monte-Carlo-based PRM (given mild assumptions). Leveraging the framework of GRPO-as-a-PRM, we identify a flaw in the GRPO objective that interacts with imbalanced process steps and rewards to hinder both exploration and exploitation (under different conditions). We propose a simple modification to the algorithm to mitigate this defect ($\lambda$-GRPO), and show that LLMs tuned with $\lambda$-GRPO outperform LLMs tuned with standard GRPO on downstream reasoning tasks—and reach peak performance more rapidly. These results show that we can leverage the hidden, built-in PRM structure within the vanilla GRPO algorithm to boost model performance without employing an explicit PRM, and with a negligible impact on training time and cost.

## 1. Introduction

Process reward models (PRMs)—models that assign reward to intermediate steps within a trajectory (see Section 2.2)—allow for finer-grained credit assignment than outcome-level signals during LLM reinforcement learning (RL), thereby yielding improved multi-step reasoning performance (Lightman et al., 2024). PRMs are therefore particularly applicable to RL training for step-by-step processes such as mathematical reasoning.

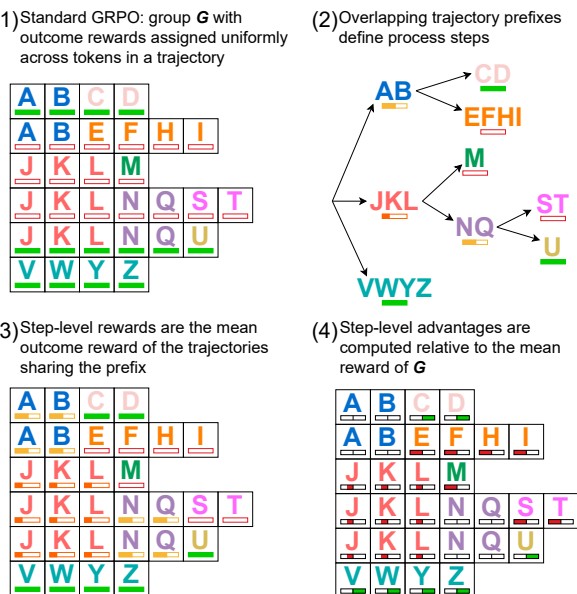

*Figure 1.* Illustration of GRPO's implicit PRM and PRM-aware advantage computation, where each row denotes a trajectory, and each box a token in (1), (3), and (4). We prove that the standard GRPO algorithm with outcome-level rewards (1) performs sub-trajectory-level credit assignment in this manner (2-4), whenever trajectories within a group share overlapping prefixes.

However, training neural PRMs requires costly, step-level human annotation (Zhang et al., 2025), and such models are particularly susceptible to reward hacking (Cui et al., 2025). These shortcomings have resulted in the limited adoption of learned PRMs for RL training (Setlur et al., 2025), leading to the development of Monte-Carlo-based and other heuristic, non-neural PRMs (e.g. Wang et al., 2024; Kazemnejad et al., 2025; Hou et al., 2025).

PRMs are typically employed with RL algorithms that employ a critic model and/or generalized advantage estimation (GAE), such as Proximal Policy Optimization (PPO; Schulman et al., 2017). Group Relative Policy Optimization (GRPO; Shao et al., 2024) eliminates the critic model and GAE of PPO: this greatly simplifies and reduces the memory consumption of RL training, which has led to the adoption of GRPO for a wide range of applications, including tool use (Qian et al., 2025; Sullivan et al., 2025), RLHF (Yang et al., 2025a), and, in particular, mathematical reasoning

---

[1]Department of Language Science and Technology, Saarland Informatics Campus, Saarland University, Saarbrücken, Germany. Correspondence to: Michael Sullivan <msullivan@lst.uni-saarland.de>.

*Proceedings of the 43rd International Conference on Machine Learning*, Seoul, South Korea. PMLR 306, 2026. Copyright 2026 by the author(s).

(Shao et al., 2024; DeepSeek-AI, 2025). However, due to its lack of GAE and critic model, GRPO has not been widely used with PRMs and, to the best of our knowledge, Shao et al. (2024), Yang et al. (2025b), Feng et al. (2025), and Ji et al. (2025) are the only instances in which GRPO is employed with step-level rewards—and these approaches necessitate the modification of the algorithm to accommodate finer-grained reward signals.

In this paper, we show that—under certain mild assumptions—GRPO is in fact equivalent to a process-reward-sensitive RL objective equipped with a Monte-Carlo-based PRM (Section 3). Specifically, we prove theoretically that GRPO assigns step-level rewards (and advantages) derived from outcome-level rewards and Monte-Carlo-sampled completions to sub-trajectories, whenever subsets of trajectories within each group share identical prefixes (see Figure 1). We then show empirically that this identical-prefix condition is often met under real-world conditions, yielding rich step-level process reward structures. These two findings definitively demonstrate that the GRPO objective covertly assigns and optimizes for complex, structured step-level rewards and advantages.

An investigation into GRPO's hidden PRM then reveals a defect in the objective function (Section 4): namely, GRPO's advantage computation interacts with imbalanced process step frequencies and process rewards to hinder both exploration and exploitation (under different conditions). For example, the first three tokens (*JKL*) of the trajectory *JKLNQU* in Figure 1 are assigned *negative* advantage—and only the last (*U*) receives positive advantage—despite the fact that this trajectory has high outcome-level reward. Optimizing for the GRPO objective will then decrease the likelihood of generating (exploiting) this high-reward trajectory in subsequent training steps (see Section 2.1).

We then propose including a PRM-aware normalization factor into the GRPO loss function ($\lambda$-GRPO): using a synthetic task, we show that our approach improves the robustness of GRPO to the effects of process step frequency imbalance. On downstream reasoning benchmarks, $\lambda$-GRPO consistently improves over standard GRPO, demonstrating the superiority of our method (Section 5). We additionally show that $\lambda$-GRPO results in a $\sim$2x training speedup and higher validation accuracy over standard GRPO. These findings suggest that future work can benefit from exploiting the implicit, step-level reward signal available to the GRPO algorithm, reducing the need for costly PRMs.

## 2. Background

### 2.1. GRPO

GRPO is a variant of PPO that discards the critic model and GAE of the latter. Instead, for each query (prompt) $x$

in the training set, GRPO nondeterministically samples a *group* $\mathbb{G}$ of trajectories (completions to $x$) and computes the advantage $a_i$ for the completion $y^{(i)} \in \mathbb{G}$ relative to the mean reward of $\mathbb{G}$, as in Equation 1, where $r^{(i)}$ denotes the reward for $y^{(i)} \in \mathbb{G}$.

$$a_i = \frac{r^{(i)} - r_{mean}(\mathbb{G})}{r_{std}(\mathbb{G})} \tag{1}$$

For $\mu \geq 1$ update iterations, GRPO optimizes the policy LLM $\pi_\theta$ to maximize the objective in Equation 2a, where $P_{i,t}$ and $D_{i,t}$ denote the token probability ratio and KL penalty, respectively (Equations 2b and 2c), and $\epsilon > 0, \beta \geq 0$ are hyperparameters.

$$L_{GRPO}(\mathbb{G}) =$$
$$\sum_{y^{(i)} \in \mathbb{G}} \frac{1}{len(y^{(i)})} \sum_{t=0}^{len(y^{(i)})-1} min\left(P_{i,t} \cdot a_i, clip_{1-\epsilon}^{1+\epsilon}(P_{i,t}) \cdot a_i\right) - D_{i,t} \tag{2a}$$

$$P_{i,t} = \frac{\pi_\theta(y_t^{(i)} \mid x, y_{:t}^{(i)})}{\pi_{\theta_{old}}(y_t^{(i)} \mid x, y_{:t}^{(i)})} \tag{2b}$$

$$D_{i,t} = \beta \cdot \left(\frac{\pi_{\theta_{ref}}(y_t^{(i)} \mid x, y_{:t}^{(i)})}{\pi_\theta(y_t^{(i)} \mid x, y_{:t}^{(i)})} - ln\frac{\pi_{\theta_{ref}}(y_t^{(i)} \mid x, y_{:t}^{(i)})}{\pi_\theta(y_t^{(i)} \mid x, y_{:t}^{(i)})} - 1\right) \tag{2c}$$

Given a trajectory $y^{(i)}$ with reward $r^{(i)}$: if $r^{(i)} < r_{mean}(\mathbb{G})$, then $a_i < 0$, and optimizing for Equation 2a has the effect of decreasing the likelihood $\pi_\theta(y^{(i)} \mid x)$. Conversely, if $r^{(i)} > r_{mean}(\mathbb{G})$, then $a_i > 0$, and so $\pi_\theta(y^{(i)} \mid x)$ is increased.

In our theoretical analysis in Section 3, we make two key assumptions: first, we assume the use of the DAPO token-level policy gradient objective (Yu et al., 2025), rather than sample-level loss. Although it differs from the original GRPO formulation laid out in Shao et al. (2024), Yu et al. (2025) show that this objective leads to more stable training, and it is the standard GRPO loss function employed in commonly used RL packages (e.g. the TRL GRPO trainer[1]).

Second, we assume that the hyperparameter $\mu$ is set to $\mu = 1$ update iteration per batch. Under this assumption, the ratio $P_{i,t}$ is fixed at 1.0 (see Equation 2b), allowing us to ignore the clipping factor of the GRPO loss function in our theoretical analysis.

---

[1]https://huggingface.co/docs/trl/main/en/grpo_trainer

Under these two assumptions, the per-group GRPO loss $L_{\text{GRPO}}(\mathbb{G})$ reduces to that in Equation 3.

$$L_{\text{GRPO}}(\mathbb{G}) = \frac{1}{\sum_{y^{(i)} \in \mathbb{G}} len(y^{(i)})} \sum_{y^{(i)} \in \mathbb{G}} \sum_{t=0}^{len(y^{(i)})-1} (P_{i,t} \cdot a_i) - D_{i,t} \tag{3}$$

### 2.2. Process Reward Models (PRMs)

Given an alphabet $\Sigma$ (i.e. set of tokens), we formally define a PRM as in Definition 1, where $y_{j:k}^{(i)}$ denotes the subsequence of $y^{(i)}$ spanning the $j^{th}$ (inclusive) to $k^{th}$ (exclusive) tokens.

**Definition 1** (Process Reward Model). A function $f_\phi \colon \Sigma^* \to (\Sigma^* \times \mathbb{R})^*$ parameterized by $\phi$ that maps a trajectory $y^{(i)} \in \Sigma^*$ to the sequence $f_\phi(y^{(i)}) = ((y_{:k_1}^{(i)}, r_0^{(i)}), (y_{k_1:k_2}^{(i)}, r_1^{(i)}), \ldots, (y_{k_n:}^{(i)}, r_n^{(i)}))$ of pairs of *process steps* (sub-trajectories) $y_{k_m:k_{m+1}}^{(i)}$ and step-level rewards $r_m^{(i)}$.

While PRMs are typically contrasted with *outcome* reward models (ORMs)—which assign a single reward to the entire trajectory—under the above definition, an ORM $f_{\phi'}'$ is simply a *trivial* PRM: i.e. $f_{\phi'}'(y^{(i)}) = ((y^{(i)}, r^{(i)}))$.

Both the division of the trajectory $g$ into steps and the assignment of rewards to those steps are dependent upon the PRM in question. When individual steps are clearly delineated—e.g. via ReAct-style prompting (Yao et al., 2023) or instructing the model to divide its reasoning into demarcated steps—the PRM can simply be directed to assign a reward to each pre-defined step (e.g. Li & Li, 2025). In other cases, trajectories are heuristically split into steps—for example, at high-entropy tokens (e.g. Hou et al., 2025).

Although the assignment of step-level reward can be performed by a model with learned parameters $\phi$ (e.g. Uesato et al., 2022), Kazemnejad et al. (2025) and Hou et al. (2025) combine Monte Carlo estimation with outcome-level rewards to yield heuristic PRMs that do not require the labor-intensive annotation of—and are less susceptible to reward-hacking than—their learned counterparts. In cases such as these in which the PRM $f_\phi$ is not learned, we simply consider $\phi$ to be fixed/trivial.

## 3. GRPO's Hidden PRM

In Section 3.1, we prove that GRPO is equivalent to a PRM-sensitive RL objective equipped with a heuristic, Monte-Carlo-based PRM. As illustrated in Figure 1, overlapping trajectory prefixes within a group define process steps, and this implicit PRM computes step-level rewards as the mean outcome-level reward of the trajectories that begin with the prefix in question.

However, this PRM is only non-trivial—i.e. not equivalent to an ORM—if subsets of trajectories within each group actually share identical prefix sub-trajectories. In Section 3.2, we empirically demonstrate that such rich, overlapping prefix structures arise very frequently under real-world conditions: this shows that GRPO "secretly" contains a non-trivial PRM.

### 3.1. Theoretical Analysis

Given a group $\mathbb{G} = \{y^{(1)}, \ldots, y^{(|\mathbb{G}|)}\}$ with outcome rewards $r^{(1)}, \ldots, r^{(|\mathbb{G}|)}$, we construct: (i) a PRM that derives process rewards from Monte Carlo estimates of outcome-level reward (Equation 5); and (ii) a PRM-sensitive RL objective, $L_{PRM}$, that is similar to GRPO (Equation 6). We then prove in Theorem 1 that for any $\mathbb{G}$, $L_{PRM}(\mathbb{G})$ is equal to $L_{GRPO}(\mathbb{G})$ as defined in Equation 3—i.e. that GRPO with outcome-level rewards is (equivalent to) a PRM-sensitive RL algorithm equipped with a Monte-Carlo-based PRM.

**Process Steps.** Let $\mathcal{B}(\mathbb{G}) = \{\lambda \subseteq \mathbb{G} \mid \exists n \geq 0 \forall y^{(i)}, y^{(k)} \in \lambda \colon y_{:n}^{(i)} = y_{:n}^{(k)}\}$ be the set of all *process sets* of $\mathbb{G}$: sets $\lambda \subseteq \mathbb{G}$ of completions such that all $y^{(i)} \in \lambda$ are identical up to the $n^{th}$ token, for some $n \geq 0$—i.e. such that all $y^{(i)} \in \lambda$ share the prefix $y_{:n}^{(i)}$. There is a natural tree structure on $\mathcal{B}(\mathbb{G})$ induced by the $\supseteq$ relation (see Figure 2).

Each $\lambda \in \mathcal{B}(\mathbb{G})$ defines a process step within each trajectory $y^{(i)} \in \lambda$, spanning the subsequence $y_{s(\lambda):e(\lambda)}^{(i)}$ from the $s(\lambda)^{th}$ to the $e(\lambda)^{th}$ tokens of $y^{(i)}$. The endpoint $e(\lambda)$ is defined as the largest $n$ such that $y_{:n}^{(i)} = y_{:n}^{(k)}$ for all $y^{(i)}, y^{(k)} \in \lambda$, and the starting point $s(\lambda)$ is defined as the endpoint of the immediate parent $Pa_{\mathcal{B}(\mathbb{G})}(\lambda)$ of $\lambda$ in the tree structure induced on $\mathcal{B}(\mathbb{G})$ (Equation 4).

$$e(\lambda) = max\{n \geq 0 \mid \forall y^{(i)}, y^{(k)} \in \lambda \colon y_{:n}^{(i)} = y_{:n}^{(k)}\}$$
$$s(\lambda) = \begin{cases} 0 & \text{if } \lambda = \mathbb{G} \\ e(Pa_{\mathcal{B}(\mathbb{G})}(\lambda)) & \text{otherwise} \end{cases} \tag{4}$$

For example, in Figure 2, $e(\mathbb{G}) = 0$, because there is no $n > 0$ such that $y_{:n}^{(i)} = y_{:n}^{(k)}$ for all $y^{(i)}, y^{(k)} \in \mathbb{G}$. As $\{y^{(3)}, y^{(4)}, y^{(5)}\}$ is a child of $\mathbb{G}$ in $\mathcal{B}(\mathbb{G})$, $s(\{y^{(3)}, y^{(4)}, y^{(5)}\}) = e(\mathbb{G}) = 0$ by definition (Equation 4). The greatest $n$ such that $y_{:n}^{(3)} = y_{:n}^{(4)} = y_{:n}^{(5)}$ is $n = 3$, because they share the prefix *JKL*, where *J*, *K*, and *L* each denote a token. Therefore, $e(\{y^{(3)}, y^{(4)}, y^{(5)}\}) = 3$: the process step defined by the set $\{y^{(3)}, y^{(4)}, y^{(5)}\}$ spans the shared subtrajectory $y_{0:3}^{(3)} = y_{0:3}^{(4)} = y_{0:3}^{(5)} = JKL$.

Similarly, the greatest $n$ such that $y_{:n}^{(4)} = y_{:n}^{(5)}$ is $n = 5$, and so $e(\{y^{(4)}, y^{(5)}\}) = 5$. As $\{y^{(4)}, y^{(5)}\}$ is a child of $\{y^{(3)}, y^{(4)}, y^{(5)}\}$ in $\mathcal{B}(\mathbb{G})$, $s(\{y^{(4)}, y^{(5)}\}) =$

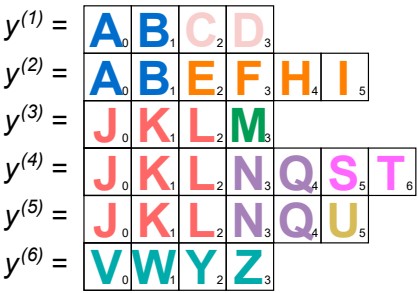 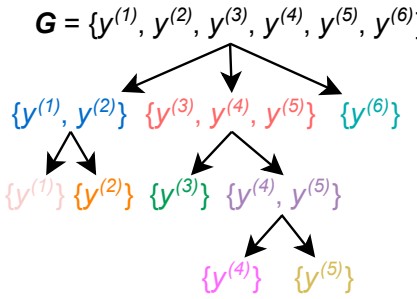

*Figure 2.* Toy example of a group $\mathbb{G} = \{y^{(1)}, \ldots, y^{(6)}\}$ (left) and its corresponding $\mathcal{B}(\mathbb{G})$ tree (right). Letters/boxes on the left-hand side denote tokens, and subscripted numbers within boxes indicate token position. Each process *set* (node in the $\mathcal{B}(\mathbb{G})$ tree) is a set of trajectories that share a common prefix, and corresponds to a process *step* (subtrajectory) spanning those shared tokens—in this figure, colored nodes in $\mathcal{B}(\mathbb{G})$ correspond to those subsequences (i.e. *process steps*) in $\mathbb{G}$ that span tokens/boxes of the same color.

$e(\{y^{(3)}, y^{(4)}, y^{(5)}\}) = 3$ by definition (Equation 4). The process step defined by $\{y^{(4)}, y^{(5)}\}$ therefore spans the shared subtrajectory $y^{(4)}_{3:5} = y^{(5)}_{3:5} = NQ$.

**Process Reward Model.** Now, consider the function $f$ that maps each $y^{(i)} \in \mathbb{G}$ to $f(y^{(i)})$ as in Equation 5, where $\lambda_0 = \mathbb{G}$, $\lambda_n = \{y^{(i)}\}$, $\lambda_0 \to \lambda_1 \to \cdots \to \lambda_n$ is the unique path from the root $\mathbb{G}$ to the leaf node $\{y^{(i)}\}$ in $\mathcal{B}(\mathbb{G})$, and $r_{mean}(\lambda_k) = mean(\{r^{(i)}\}_{y^{(i)} \in \lambda_k})$ is the mean outcome reward of the trajectories in $\lambda_k$.

$$f(y^{(i)}) = ((y^{(i)}_{s(\lambda_0):e(\lambda_0)}, r_{mean}(\lambda_0)),$$
$$(y^{(i)}_{s(\lambda_1):e(\lambda_1)}, r_{mean}(\lambda_1)),$$
$$\ldots, \tag{5}$$
$$(y^{(i)}_{s(\lambda_n):e(\lambda_n)}, r_{mean}(\lambda_n)))$$

For each $\lambda_k$ along the path $\lambda_0 \to \lambda_1 \to \cdots \to \lambda_n$, $f$ assigns the Monte-Carlo-sampled process reward $r_{mean}(\lambda_k)$ to the subtrajectory $y^{(i)}_{s(\lambda_k):e(\lambda_k)}$ corresponding to $\lambda_k$. Clearly, $f$ is a PRM as defined in Definition 1.

For example, in Figure 2, the sub-trajectory $y^{(4)} = JKLNQST$ is divided into three process steps: $y^{(4)}_{:3} = JKL$, $y^{(4)}_{3:5} = NQ$, and $y^{(4)}_{5:} = ST$. Each step corresponds to a shared prefix, and the corresponding step-level reward is computed as the mean reward of the trajectories that share that prefix—i.e. as the Monte Carlo estimate of the outcome reward for a completion to the prefix. This is to say that the step-level reward for $y^{(4)}_{:3}$ is $r_{mean}(\{y^{(3)}, y^{(4)}, y^{(5)}\})$: the mean reward of each trajectory that begins with $JKL$. The step-level reward for $y^{(4)}_{3:5}$ is the mean reward of each trajectory that begins with $JKLNQ$—i.e. $r_{mean}(\{y^{(4)}, y^{(5)}\})$. The step-level reward for the final subsequence $y^{(4)}_{5:}$—which does not overlap with any other trajectories—is simply the outcome-level reward for $y^{(4)}$: $r_{mean}(\{y^{(4)}\}) = r^{(4)}$.

**PRM-Aware RL Objective.** We incorporate the PRM $f$ of Equation 5 into the PRM-aware RL objective below via the token-level reward $R_{i,t}$, which uniformly assigns the reward $R_{i,t} = r_{mean}(\lambda_k)$ to each token $y^{(i)}_t$ in the span $y^{(i)}_{s(\lambda_k):e(\lambda_k)}$, for each such $(y^{(i)}_{s(\lambda_k):e(\lambda_k)}, r_{mean}(\lambda_k))$ pair in $f(y^{(i)})$ (see Figure 1).

Now, we define the token-level advantage $A_{i,t}$ for the token $y^{(i)}_t$ in an analogous manner to the actual GRPO definition in Equation 1—i.e. as the normalized difference between the token-level reward $R_{i,t}$ for $y^{(i)}_t$ and the mean outcome reward of $\mathbb{G}$: $A_{i,t} = (R_{i,t} - r_{mean}(\mathbb{G}))/r_{std}(\mathbb{G})$.

Replacing the term $a_i$ with $A_{i,t}$ in Equation 3 yields a PRM-aware RL objective (Equation 6).

$$L_{\text{PRM}}(\mathbb{G}) = \frac{1}{\sum_{y^{(i)} \in \mathbb{G}} len(y^{(i)})} \sum_{y^{(i)} \in \mathbb{G}} \sum_{t=0}^{len(y^{(i)})-1} (P_{i,t} \cdot A_{i,t}) - D_{i,t} \tag{6}$$

**Objective Equivalence between $L_{GRPO}$ and $L_{PRM}$.** We now prove in Theorem 1 that the standard GRPO objective defined in Equation 3 with outcome-level rewards ($L_{\text{GRPO}}$) is equivalent to the PRM-aware RL objective defined in Equation 6 equipped with the Monte-Carlo-based PRM defined in Equation 5 ($L_{\text{PRM}}$).

**Theorem 1.** *For any query $x$, policy $\pi_\theta$, and group $\mathbb{G} \sim \pi_\theta(- \mid x)$ with outcome-level rewards $\{r^{(i)}\}_{y^{(i)} \in \mathbb{G}}$: $L_{GRPO}(\mathbb{G}) = L_{PRM}(\mathbb{G})$.*

*Proof.* We first prove that, for any process set $\lambda \in \mathcal{B}(\mathbb{G})$ and any $t$ such that $s(\lambda) \le t < e(\lambda)$, the sum over each $y^{(i)} \in \lambda$ of the PRM loss terms $P_{i,t} \cdot A_{i,t} - D_{i,t}$ is equivalent to the sum of the standard GRPO loss terms $P_{i,t} \cdot a_i - D_{i,t}$ (Lemma 1).

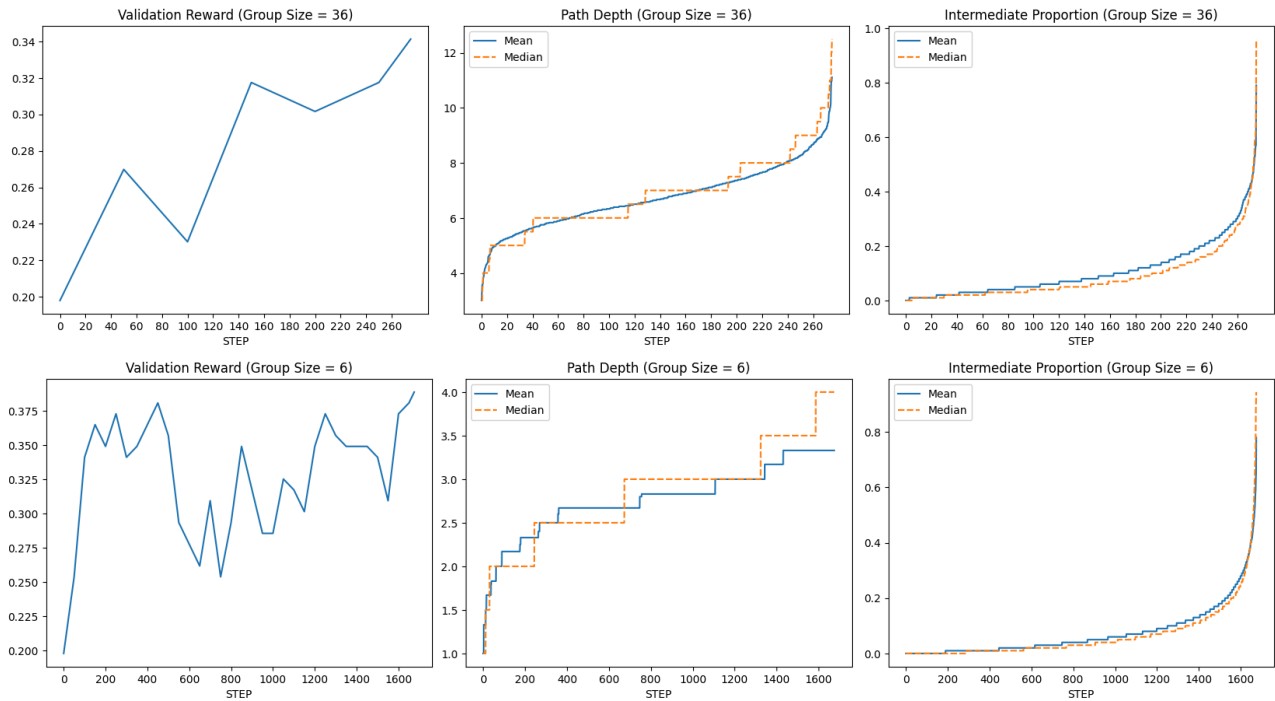

*Figure 3.* Validation reward (exact-match accuracy; left), $\mathcal{B}(\mathbb{G})$ root-to-terminal path depth (center), and proportions of trajectories spanned by intermediate (non-terminal) process steps (right) for GRPO runs with group sizes of 36 (top) and 6 (bottom).

**Lemma 1.** *For any $\lambda \in \mathcal{B}(\mathbb{G})$ and any integer $t$ such that $s(\lambda) \leq t < e(\lambda)$:*

$$\sum_{y^{(i)} \in \lambda} \left( P_{i,t} \cdot a_i \right) - D_{i,t} = \sum_{y^{(i)} \in \lambda} \left( P_{i,t} \cdot A_{i,t} \right) - D_{i,t}$$

*Proof.* Appendix B. □

Now, let $t_{max} = max_{y^{(i)} \in \mathbb{G}} \, len(y^{(i)})$. For each $0 \leq t < t_{max}$, we can define a partition $\mathbb{X}_t \subseteq \mathcal{B}(\mathbb{G})$ of $\{y^{(i)} \in \mathbb{G} \mid len(y^{(i)}) \leq t\}$ such that $\mathbb{X}_t = \{\lambda \in \mathcal{B}(\mathbb{G}) \mid s(\lambda) \leq t < e(\lambda)\}$ is the set of all process sets corresponding to a token span containing the index $t$. The GRPO loss term $L_{\text{GRPO}}(\mathbb{G})$ (Equation 3) can be equivalently expressed as in Equation 7 (and analogously for $L_{\text{PRM}}(\mathbb{G})$ of Equation 6).

$$L_{\text{GRPO}}(\mathbb{G}) = \frac{1}{\sum_{y^{(i)} \in \mathbb{G}} len(y^{(i)})} \cdot \sum_{t=0}^{t_{max}-1} \sum_{\lambda \in \mathbb{X}_t} \sum_{y^{(i)} \in \lambda} \left( P_{i,t} \cdot a_i \right) - D_{i,t} \tag{7}$$

We then have the following equalities by Lemma 1 and Equation 7:

$$
\begin{aligned}
L_{\text{GRPO}}(\mathbb{G}) &= \frac{1}{\sum_{y^{(i)} \in \mathbb{G}} len(y^{(i)})} \cdot \sum_{y^{(i)} \in \mathbb{G}} \sum_{t=0}^{len(y^{(i)})-1} \left( P_{i,t} \cdot a_i \right) - D_{i,t} \\
&= \frac{1}{\sum_{y^{(i)} \in \mathbb{G}} len(y^{(i)})} \cdot \sum_{t=0}^{t_{max}-1} \sum_{\lambda \in \mathbb{X}_t} \sum_{y^{(i)} \in \lambda} \left( P_{i,t} \cdot a_i \right) - D_{i,t} \\
&= \frac{1}{\sum_{y^{(i)} \in \mathbb{G}} len(y^{(i)})} \cdot \sum_{t=0}^{t_{max}-1} \sum_{\lambda \in \mathbb{X}_t} \sum_{y^{(i)} \in \lambda} \left( P_{i,t} \cdot A_{i,t} \right) - D_{i,t} \\
&= \frac{1}{\sum_{y^{(i)} \in \mathbb{G}} len(y^{(i)})} \cdot \sum_{y^{(i)} \in \mathbb{G}} \sum_{t=0}^{len(y^{(i)})-1} \left( P_{i,t} \cdot A_{i,t} \right) - D_{i,t} = L_{\text{PRM}}(\mathbb{G})
\end{aligned}
$$

□

### 3.2. Empirical Analysis

The theoretical analysis in Section 3.1 shows that the GRPO objective is equivalent to a PRM-aware objective with a Monte Carlo PRM: although GRPO appears to assign credit uniformly to each token within a trajectory, there are in fact—theoretically—finer-grained reward dynamics at play.

However, if the prefixes of the trajectories within a group do not overlap with one another in practice, then this implicit PRM is trivial—i.e. equivalent to an ORM. Similarly, if there is only a small degree of overlap between trajectories, then—while not technically trivial—the vast majority of tokens will still receive purely outcome-level reward signal

during GRPO training. It therefore remains to be shown empirically that trajectory prefixes overlap substantially in a real-world setting.

To analyze the step-level reward structure that occurs in real-world GRPO training, we computed the $\mathcal{B}(\mathbb{G})$ tree for each group $\mathbb{G}$ generated during training, and calculated two metrics reflecting the complexity and size of the underlying process steps: *path depth* and *intermediate proportion*.

1. **Path Depth:** the number of nodes along the path from a terminal node $\{y^{(i)}\}$ to the root $\mathbb{G}$. This metric is a proxy for the complexity of the $\mathcal{B}(\mathbb{G})$ structure: if trajectory prefixes do not overlap, then the induced tree is flat—each terminal $\{y^{(i)}\}$ is directly connected to the root $\mathbb{G}$—and so the path depth is zero. Conversely, higher path depths reflect richer prefix overlap.

2. **Intermediate Proportion:** the proportion of tokens in a trajectory $y^{(i)}$ that are contained within a prefix that overlaps with one or more other trajectories in $\mathbb{G}$. This reflects the relative size of the induced process steps: higher intermediate proportions indicate that a greater proportion of the tokens in $y^{(i)}$ belong to an intermediate process step and are therefore assigned non-trivial process-level reward.

**Experimental Setup.** We fine-tuned two DeepSeek-R1-Distill-Qwen-1.5B models (DeepSeek-AI, 2025) on the OpenRS (Dang & Ngo, 2025) math-reasoning dataset using the standard GRPO objective of Equation 3. We selected 125 OpenRS examples at random as a validation set.

The first model trained for 1675 steps with a group size of six and a learning rate of $6 \times 10^{-6}$. The second was trained with a group size of 36 and a learning rate of $10^{-6}$ for 275 steps (due to the larger group size). Both models were trained with a maximum new token limit of 4096, a batch size of four, and a temperature of 0.75. Additional training details are located in Appendix C.

**Results.** Figure 3 shows that both path depth and intermediate proportion increase drastically as validation reward saturates, for group sizes of six and 36: this indicates that increasingly rich process step structures arise as the model converges on a locally optimal policy. These results are supported by Yu et al. (2025), who find that entropy decreases sharply as GRPO training progresses.

In addition, only twelve of 6,700 $\mathcal{B}(\mathbb{G})$ trees were flat with a group size of six ($\sim$0.2%): 99.8% of the generated groups induced non-trivial process rewards. With a group size of 36, zero flat $\mathcal{B}(\mathbb{G})$ trees arose out of the 1,100 generated groups. Examples of non-trivial $\mathcal{B}(\mathbb{G})$ structures from this experiment are given in Appendix E.

## 4. Proposed Approach: $\lambda$-GRPO

The theoretical and empirical analyses in Section 3 demonstrate that GRPO's underlying PRM is non-trivial under real-world conditions. In this section, we first show that this PRM carries a flaw that is detrimental to RL training (Section 4.1). We then propose a minor modification to the GRPO algorithm (Section 4.2), and demonstrate empirically that our approach mitigates this shortcoming (Section 4.3).

### 4.1. Problem Statement

**Preliminaries.** First, recall that for any $\lambda \in \mathcal{B}(\mathbb{G})$, $t$ such that $s(\lambda) \leq t < e(\lambda)$, and $y^{(i)}, y^{(k)} \in \lambda$: $y^{(i)}_{:t+1} = y^{(k)}_{:t+1}$ by definition, and so $P_{i,t} = P_{k,t}$ and $D_{i,t} = D_{k,t}$ (Equations 2b-2c). Again by definition, $y^{(i)}, y^{(k)} \in \lambda$ implies that $R_{i,t} = R_{k,t} = r_{mean}(\lambda)$, and so $A_{i,t} = A_{k,t}$. As such, we may define $\hat{P}_t(\lambda) = P_{k,t}$, $\hat{D}_t(\lambda) = D_{k,t}$, and $\hat{A}(\lambda) = A_{k,t}$ below, choosing any arbitrary $y^{(k)} \in \lambda$.

Now, for each $y^{(i)} \in \mathbb{G}$ and each $0 \leq t < len(y^{(i)})$, let $\lambda^{(i,t)} \in \mathcal{B}(\mathbb{G})$ denote the unique process set such that $y^{(i)} \in \lambda^{(i,t)}$ and $s(\lambda^{(i,t)}) \leq t < e(\lambda^{(i,t)})$: $\lambda^{(i,t)}$ is the process step to which the token $y^{(i)}_t$ belongs. In Figure 2, $\lambda^{(i,t)}$ corresponds to the set on the right-hand side whose color matches that of $y^{(i)}_t$ (on the left-hand side): for example, $y^{(1)}_0 = A$ belongs to the span corresponding to $\{y^{(1)}, y^{(2)}\}$, and so $\lambda^{(1,0)} = \{y^{(1)}, y^{(2)}\}$. Similarly, $y^{(1)}_3 = D$ belongs to the span corresponding to $\{y^{(1)}\}$, and so $\lambda^{(1,3)} = \{y^{(1)}\}$.

**Imbalanced Process Step Frequency.** Viewing the GRPO objective in terms of process set partitions $\mathbb{X}_t$ (see Equation 7), we note that the contribution of each trajectory $y^{(i)} \in \mathbb{G}$ to the loss at index $t$ is identical to that of all other trajectories in the process set $\lambda^{(i,t)}$:

$$
\begin{aligned}
L_{GRPO}(\mathbb{G}) &= \frac{1}{\sum_{y^{(i)} \in \mathbb{G}} len(y^{(i)})} \cdot \sum_{y^{(i)} \in \mathbb{G}} \sum_{t=0}^{len(y^{(i)})-1} (P_{i,t} \cdot a_i) - D_{i,t} \\
&= \frac{1}{\sum_{y^{(i)} \in \mathbb{G}} len(y^{(i)})} \cdot \sum_{t=0}^{t_{max}-1} \sum_{\lambda \in \mathbb{X}_t} \sum_{y^{(i)} \in \lambda} (P_{i,t} \cdot A_{i,t}) - D_{i,t} \\
&= \frac{1}{\sum_{y^{(i)} \in \mathbb{G}} len(y^{(i)})} \cdot \sum_{t=0}^{t_{max}-1} \sum_{\lambda \in \mathbb{X}_t} |\lambda| \cdot ((\hat{P}_t(\lambda) \cdot \hat{A}(\lambda)) - \hat{D}_t(\lambda))
\end{aligned}
$$
(8)

The contribution of each process set $\lambda \in \mathcal{B}(\mathbb{G})$ to the overall loss, $\hat{P}_t(\lambda) \cdot \hat{A}(\lambda) - \hat{D}_t(\lambda)$, is scaled by the frequency $|\lambda|$ across trajectories of the process step defined by $\lambda$. Consider, for example, some process set $\lambda$ with $|\lambda| \gg 1$. If $\hat{A}(\lambda) > 0$, then the increase in probability assigned to the process step corresponding to $\lambda$ by $\pi_\theta$ under GRPO is compounded by a factor of $|\lambda|$, decreasing the likelihood of exploring pro-

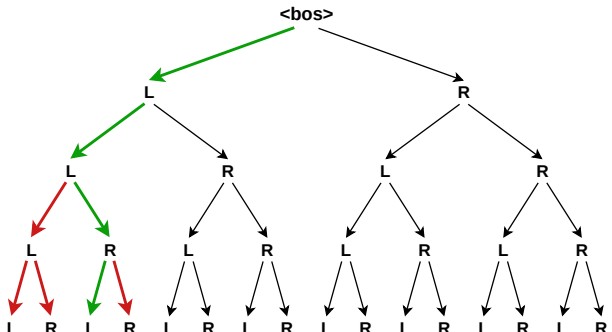

Figure 4. Example environment with $T = LLRL$ and $n = 2$. The target path $T$ (green) is assigned a reward of +1.0, and all non-$T$ paths sharing the prefix $T_{:n} = LL$ (red) are assigned a reward of $r_{neg}$. All other paths have a reward of +0.7.

cess steps that are dissimilar from $\lambda$ in subsequent training episodes, and thereby harming exploration.

Conversely, if $\hat{A}(\lambda) < 0$, then the *decrease* in probability assigned to $\lambda$ under GRPO is compounded by a factor of $|\lambda|$, decreasing the likelihood of exploiting high-reward trajectories in $\lambda$. To illustrate, consider the group $\mathbb{G}$ in Figure 2, assume $r^{(1)} = r^{(2)} = r^{(6)} = 0.5$, $r^{(4)} = r^{(5)} = 0$, $r^{(3)} = 1$, and let $\lambda = \{y^{(3)}, y^{(4)}, y^{(5)}\}$ ($\lambda$ corresponds to the process step spanning *JKL*; see Section 3.1). Then $r_{mean}(\mathbb{G}) = 0.42$ and $r_{mean}(\lambda) = 0.33$, and so $\hat{A}(\lambda) = -0.22$: despite the fact that $y^{(3)} = $ *JKLM* has the highest reward in $\mathbb{G}$, the probability of the sub-trajectory $y^{(3)}_{:3} = $ *JKL* is *decreased* under the GRPO objective, which decreases the overall likelihood of the completion $y^{(3)}$. The term $|\lambda|$ in Equation 8 then scales this decrease in probability by a factor of three.

## 4.2. $\lambda$-GRPO

We propose scaling the token-level loss for $y^{(i)}_t$ by $|\lambda^{(i,t)}|^{-1}$ ($\lambda$-GRPO; Equation 9): this has the effect of canceling out the term $|\lambda|$ in Equation 8, so that each process set contributes equally to the loss at index $t$.

$$L_{\lambda\text{-GRPO}}(\mathbb{G}) =$$

$$\frac{1}{\sum_{y^{(i)} \in \mathbb{G}} len(y^{(i)})} \cdot \sum_{y^{(i)} \in \mathbb{G}} \sum_{t=0}^{len(y^{(i)})-1} \frac{(P_{i,t} \cdot a_i) - D_{i,t}}{|\lambda^{(i,t)}|} =$$

$$\frac{1}{\sum_{y^{(i)} \in \mathbb{G}} len(y^{(i)})} \cdot \sum_{t=0}^{t_{max}-1} \sum_{\lambda \in \mathbb{X}_t} (\hat{P}_t(\lambda) \cdot \hat{A}(\lambda)) - \hat{D}_t(\lambda) \quad (9)$$

The time complexity of building $\mathcal{B}(\mathbb{G})$ and computing the per-token scaling terms $|\lambda^{(i,t)}|$ in Equation 9 is $\mathcal{O}(k^2 n)$ for a group size $k$ and maximum sequence length $n$. However,

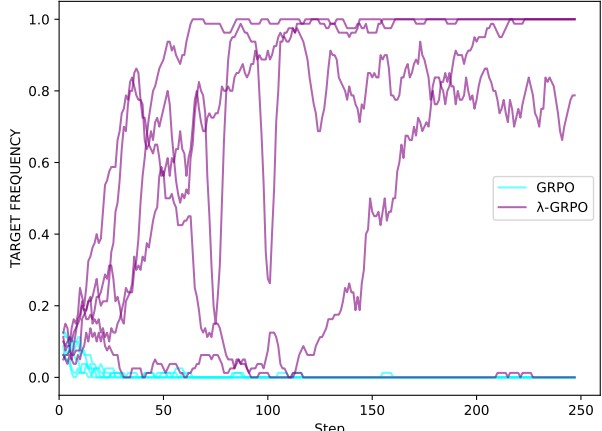

Figure 5. Per-step target path frequency for GRPO and $\lambda$-GRPO with $n = 1$ and $r_{neg} = -1.0$, plotted across all five seeds.

| $n$ | $r_{neg}$ | GRPO | $\lambda$-GRPO | $\Delta$ |
|-----|-----------|--------|----------------|----------|
|     | -0.5      | 0.3995 | 0.5663         | +0.1668  |
| 1   | -1.0      | 0.0000 | 0.7535         | +0.7535  |
|     | -1.5      | 0.2000 | 0.4463         | +0.2463  |
|     | -0.5      | 0.1980 | 0.4292         | +0.2312  |
| 2   | -1.0      | 0.0488 | 0.3612         | +0.3125  |
|     | -1.5      | 0.0000 | 0.3820         | +0.3820  |

Table 1. Target frequency within the last 50 training steps (averaged over five seeds) for GRPO and $\lambda$-GRPO across all experimental configurations: $\Delta$ denotes the row-wise difference between $\lambda$-GRPO and standard GRPO target frequencies.

the multiplicative constant hidden in the big-$\mathcal{O}$ notation is very small, and so $\lambda$-GRPO introduces negligible overhead in practice: measuring wall-clock time on the trajectories from our experiments in Section 3.2, we recorded overhead of 1.19e-7 ($k = 6$) and 1.21e-7 ($k = 36$) seconds/token on a single Intel i7 CPU—amounting to 8.38 and 10.27 seconds of total overhead (respectively) for each training run.

## 4.3. Preliminary Evaluation

In order to assess the robustness of GRPO and $\lambda$-GRPO with respect to imbalanced process step/reward frequency (see Section 4.1), we evaluated these objectives on a toy, synthetic task using GPT-2-small (Radford et al., 2018).

**Setup.** The environment for each training run consisted of a depth-four binary tree. The model traversed the tree from the root to a leaf node, taking at each step one of two possible actions (tokens): $L$ ("left") and $R$ ("right")—all other tokens were masked.

The reward dynamics of each environment were then configured to model the anti-exploration conditions hypothesized

| Model | $\beta$ | Version | AIME24 | MATH-500 | AMC23 | Minerva | OB | Avg. |
|---|---|---|---|---|---|---|---|---|
| Qwen | — | Base | $0.2000_{\pm0.0743}$ | $0.8300_{\pm0.0168}$ | $0.7500_{\pm0.0693}$ | $\mathbf{0.2978_{\pm0.0278}}$ | $0.5096_{\pm0.0193}$ | $0.5175$ |
| | 0.0 | GRPO | $0.3333_{\pm0.0875}$ | $0.7660_{\pm0.0190}$ | $0.6250_{\pm0.0775}$ | $0.2500_{\pm0.0263}$ | $0.4444_{\pm0.0191}$ | $0.4837$ |
| | | $\lambda$-GRPO (ours) | $0.3667_{\pm0.0895}$ | $0.8460_{\pm0.0162}$ | $0.7500_{\pm0.0693}$ | $0.2904_{\pm0.0276}$ | $0.5348_{\pm0.0192}$ | $0.5576$ |
| | 0.04 | GRPO | $0.3000_{\pm0.0851}$ | $\mathbf{0.8660_{\pm0.0152}}$ | $0.7500_{\pm0.0693}$ | $0.2610_{\pm0.0267}$ | $0.5200_{\pm0.0192}$ | $0.5394$ |
| | | $\lambda$-GRPO (ours) | $\mathbf{0.4000_{\pm0.0910}}$ | $0.8340_{\pm0.0167}$ | $\mathbf{0.8000_{\pm0.0641}}$ | $0.2978_{\pm0.0278}$ | $\mathbf{0.5378_{\pm0.0192}}$ | $\mathbf{0.5739}$ |
| Llama | — | Base | $0.0000_{\pm0.0000}$ | $0.2280_{\pm0.0188}$ | $0.0750_{\pm0.0422}$ | $0.0478_{\pm0.0130}$ | $0.0563_{\pm0.0089}$ | $0.0814$ |
| | 0.0 | GRPO | $0.0000_{\pm0.0000}$ | $0.2300_{\pm0.0188}$ | $0.0750_{\pm0.0422}$ | $0.0551_{\pm0.0130}$ | $0.0607_{\pm0.0089}$ | $0.0842$ |
| | | $\lambda$-GRPO (ours) | $0.0000_{\pm0.0000}$ | $0.2620_{\pm0.0197}$ | $0.1250_{\pm0.0530}$ | $0.0515_{\pm0.0134}$ | $0.0622_{\pm0.0092}$ | $0.1001$ |
| | 0.04 | GRPO | $0.0000_{\pm0.0000}$ | $0.2180_{\pm0.0185}$ | $0.1750_{\pm0.0608}$ | $0.0515_{\pm0.0134}$ | $0.0533_{\pm0.0087}$ | $0.0996$ |
| | | $\lambda$-GRPO (ours) | $0.0333_{\pm0.0333}$ | $0.2560_{\pm0.0195}$ | $0.0750_{\pm0.0422}$ | $0.0735_{\pm0.0159}$ | $0.0489_{\pm0.0083}$ | $0.0973$ |

*Table 2.* Exact-match accuracy for the base and GRPO-/$\lambda$-GRPO-trained Llama and Qwen models on downstream reasoning datasets (OB = OlympiadBench). The best results in each column are indicated in **bold**, and the best results within each model type (i.e. Llama or Qwen) are underlined. Confidence intervals are subscripted. For each $\lambda$-GRPO-trained model, results are given in green if it outperforms its GRPO-trained counterpart and the base model; yellow if it outperforms only its GRPO-trained counterpart; orange if it only improves over the base model; and red if it fails to outperform either model (see Table 3 in the Appendix for exact differences).

in Section 4.1. For each training run, we sampled one path $T$ at random—e.g. $T = LLRL$—and assigned the maximum reward (+1.0) to $T$. For a fixed $1 \leq n < 4$, we assigned a negative reward $r_{neg} < 0$ to all paths sharing the prefix $T_{:n}$ with $T$ (aside from $T$ itself). All other paths in the tree were assigned a lower positive reward of +0.7.

To illustrate the intuition behind this setup, consider the environment depicted in Figure 4: if the trajectories $T = LLRL$, $X = LLLR$, and $Y = LLRR$ are in the same group, then the mean reward of the process set $\{T, X, Y\}$ is negative, and therefore the process reward corresponding to the subtrajectory $LL$ is negative as well. Under such conditions, the anti-exploration hypothesis in Section 4.1 predicts that GRPO will struggle to converge on the maximum-reward trajectory $T$.

We evaluated this setup across a range of configurations, conducting a grid search across $n \in \{1, 2\}$, $r_{neg} \in \{-0.5, -1.0, -1.5\}$, with a learning rate[2] of $10^{-5}$ and a group size of 16. Each objective (i.e. GRPO/$\lambda$-GRPO) was evaluated across five seeds on each hyperparameter configuration with 250 training steps, and we recorded the mean proportion (across all five seeds) of occurrences of $T$ within the last 50 training steps.

**Results.** Under all hyperparameter configurations, $\lambda$-GRPO converges on the target $T$ more frequently than standard GRPO (Table 1). Viewing the per-step target frequencies across all five seeds for $n = 1$ and $r_{neg} = -1.0$ (Figure 5), we see that GRPO-trained models do in fact generate the target path during early training: they did not fail to discover $T$—rather, they failed to exploit it. For the same configuration, only one out of the five $\lambda$-GRPO models entirely fails to exploit the target path.

---

[2]We also evaluated with a learning rate of $5 \times 10^{-5}$: all configurations failed entirely to converge on $T$ under both objectives.

## 5. Experiments

To evaluate $\lambda$-GRPO under real-world conditions, we fine-tuned DeepSeek-R1-Distill-Qwen-1.5B and Llama-3.2-1B-Instruct[3] with the $\lambda$-GRPO (Equation 9) objective on the OpenRS dataset of Section 3.2, and compared them to standard GRPO (Equation 3) models tuned with an identical setup. The models were evaluated on five reasoning benchmarks (see Section 5.1), using peak performance on a withheld OpenRS validation set for checkpoint selection.

### 5.1. Setup

All models were trained for 1000 steps with a group size of six, a batch size of four, a maximum of 4096 new tokens, and a temperature of 0.75. We conducted two sets of trials across the two models (four total trials): one in which the KL coefficient $\beta = 0.0$, and a second with $\beta = 0.04$. The Qwen models were trained with a learning rate of $10^{-6}$; the Llama models were trained with a learning rate of $5 \times 10^{-7}$ for the $\beta = 0.0$ trial and $10^{-7}$ for $\beta = 0.04$ (as training was highly unstable with higher learning rates for Llama). Additional training details are located in Appendix C.

We evaluated the models on the AIME24[4], MATH-500 (Hendrycks et al., 2021; Lightman et al., 2024), AMC23[5], Minerva (Lewkowycz et al., 2022), and OlympiadBench (He et al., 2024) benchmarks, using the LightEval framework (Habib et al., 2023; Dang & Ngo, 2025). As in the experiment in Section 3.2, we withheld 125 examples from the OpenRS dataset as a validation split.

---

[3]https://huggingface.co/meta-llama/Llama-3.2-1B-Instruct
[4]https://huggingface.co/datasets/AI-MO/aimo-validation-aime
[5]https://huggingface.co/datasets/AI-MO/aimo-validation-amc

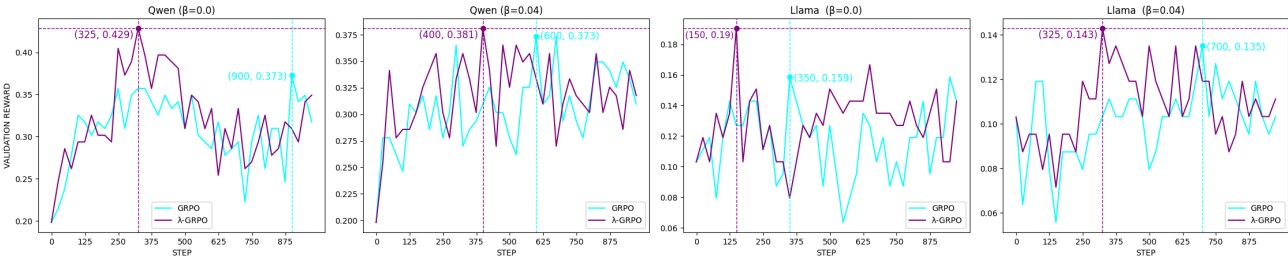

*Figure 6.* Models' validation accuracy across training steps. Peak accuracy is highlighted by vertical, dashed lines.

## 5.2. Results

In total, the $\lambda$-GRPO models outperformed standard GRPO on 15/20 cells (excluding average performance) in Table 2, and they improved over the base models on 14/20 cells. Only the Llama $\lambda$-GRPO model with $\beta = 0.04$ failed to outperform its GRPO counterpart on average downstream performance—although this model still outperformed standard GRPO on a majority (3/5) of the tasks.

In addition, all four $\lambda$-GRPO models reached a higher validation accuracy in fewer steps than their GRPO-tuned counterparts (see Figure 6): on average, $\lambda$-GRPO represents a more than 10% increase over the standard GRPO validation accuracy—in less than half of the number of training steps.

## 6. Related Work

**Monte Carlo Sampling for Finer-Grained Rewards.** The labor-intensive annotation required to obtain step-level rewards for PRM training has driven the development of heuristic methods for finer-grained reward signals, such as those based on Monte Carlo estimation. For example, Kazemnejad et al. (2025) replace the critic model in PPO with Monte Carlo estimation of outcome reward, while Wang et al. (2024) use Monte Carlo estimation to obtain step-level PRM training rewards.

Xie et al. (2024) generate step-level preference data for Direct Preference Optimization (DPO; Rafailov et al., 2023) training via Monte Carlo Tree Search and outcome-level rewards. Similarly, Hou et al. (2025) split generated trajectories at high-entropy tokens to construct $\mathcal{B}(\mathbb{G})$-like trees, which are then used to derive subtrajectory-level rewards. Yang et al. (2025b) and Ji et al. (2025) use analogous methods to generate process rewards for GRPO-like training. Pan et al. (2026) propose a similar GRPO-like RL algorithm for ranking systems that perturbs outputs post-generation to estimate step-level credit from outcome-level reward.

These methods are orthogonal to ours: they use Monte Carlo estimation to explicitly derive step-level rewards from outcome-level rewards, while we leverage the implicit step-level rewards already present in standard GRPO.

**PRMs with GRPO.** Shao et al. (2024) modify the advantage computation of GRPO to account for step-level rewards. In contrast, Feng et al. (2025) construct a two-level variant of GRPO, in which standard, trajectory-level GRPO advantage is combined with a novel, step-level GRPO advantage. Our results in Sections 3 and 5 call into question the need to adapt the objective to step-level rewards, given the rich step-level reward signal already present in standard GRPO.

**Connections between PRMs and Outcome-level Reward.** Rafailov et al. (2024) prove that DPO can learn any token-level reward function—expressed as the difference in conditional log probability between the policy and reference models—given an appropriate training dataset. We prove that GRPO with an ORM $r$ is equivalent to a PRM-sensitive RL algorithm with a PRM whose process rewards are given by an on-policy Monte Carlo estimate of the expected reward under $r$ for the process step in question.

## 7. Conclusion

In this paper, we demonstrated both theoretically and empirically that the standard GRPO algorithm is equivalent to a PRM-aware RL objective equipped with a PRM that derives step-level rewards via Monte Carlo estimation of expected outcome reward. We then showed that the advantage computation in this hidden PRM is vulnerable to imbalanced process step frequency and process rewards, and as a result is potentially detrimental to exploration and exploitation.

To mitigate this flaw, we introduced $\lambda$-GRPO, which adds a process-step-aware scaling factor to the GRPO objective. Models fine-tuned with $\lambda$-GRPO improve over standard GRPO on downstream reasoning tasks, and reach peak performance in half the number of training steps.

These results indicate that it is possible to leverage the existing PRM structure inherent in the standard GRPO algorithm, rather than employing costly, explicitly defined PRMs. The limitations of this work are discussed in Appendix A.

We release all code used in our experiments on GitHub[6].

---

[6]https://github.com/coli-saar/grpo-prm

## Acknowledgments

We gratefully acknowledge the stimulating research environment of the GRK 2853/1 "Neuroexplicit Models of Language, Vision, and Action", funded by the Deutsche Forschungsgemeinschaft (DFG; German Research Foundation) under project number 471607914.

## Impact Statement

This paper presents work whose goal is to advance the field of machine learning. There are many potential societal consequences of our work, none of which we feel must be specifically highlighted here.

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

## A. Limitations

Due to computational resource constraints, we were only able to conduct the experiments in Sections 3.2 and 5 with relatively small models: 1.5 billion (Qwen) and 1 billion (Llama) parameters. Similarly, we only use one dataset for RL training in both experiments—although OpenRS is a combination of the s1 (Muennighoff et al., 2025) and DeepScaleR (Luo et al., 2025) datasets. Future work should extend our findings regarding the non-triviality of the GRPO-induced PRM and the effectiveness of $\lambda$-GRPO to larger models and more diverse (training) datasets.

Finally, the objective of this work is to expose the PRM induced by the GRPO algorithm, and to highlight the deficiencies of that PRM as described in Section 4. To that end, our proposed $\lambda$-GRPO method only fully remedies the anti-exploration effect of the GRPO-induced PRM: the impact of the anti-exploitation effect is merely lessened by our approach. In future work, we intend to investigate more extensive modifications to the GRPO algorithm, with the goal of entirely solving the problems laid out in Section 4.1.

## B. Proof of Lemma 1

*Proof.* As discussed in Section 4.1, recall that for any $y^{(i)}, y^{(k)} \in \lambda$, $y^{(i)}_{:t+1} = y^{(k)}_{:t+1}$ by definition: therefore, $P_{i,t} = P_{k,t}$ and $D_{i,t} = D_{k,t}$ (Equations 2b-2c). Again by definition, $y^{(i)}, y^{(k)} \in \lambda$ implies that $R_{i,t} = R_{k,t} = r_{mean}(\lambda)$, and so $A_{i,t} = A_{k,t}$. As such, we may define $\hat{P}_t(\lambda) = P_{k,t}$, $\hat{D}_t(\lambda) = D_{k,t}$, and $\hat{A}(\lambda) = A_{k,t}$, choosing any arbitrary $y^{(k)} \in \lambda$. We then have the following equivalences:

$$\sum_{y^{(i)} \in \lambda} (P_{i,t} \cdot A_{i,t}) - D_{i,t}$$

$$= |\lambda| \cdot ((\hat{P}_t(\lambda) \cdot \hat{A}(\lambda)) - \hat{D}_t(\lambda))$$

$$= |\lambda| \cdot \left( \left( \hat{P}_t(\lambda) \cdot \frac{r_{mean}(\lambda) - r_{mean}(\mathbb{G})}{r_{std}(\mathbb{G})} \right) - \hat{D}_t(\lambda) \right)$$

$$= |\lambda| \cdot \left( \left( \hat{P}_t(\lambda) \cdot \frac{\sum_{y^{(i)} \in \lambda} \frac{r^{(i)}}{|\lambda|} - r_{mean}(\mathbb{G})}{r_{std}(\mathbb{G})} \right) - \hat{D}_t(\lambda) \right)$$

$$= \left( \hat{P}_t(\lambda) \frac{|\lambda|(\sum_{y^{(i)} \in \lambda} \frac{r^{(i)}}{|\lambda|} - r_{mean}(\mathbb{G}))}{r_{std}(\mathbb{G})} \right) - |\lambda| \hat{D}_t(\lambda)$$

$$= \left( \hat{P}_t(\lambda) \frac{\sum_{y^{(i)} \in \lambda} r^{(i)} - \sum_{y^{(i)} \in \lambda} r_{mean}(\mathbb{G})}{r_{std}(\mathbb{G})} \right) - |\lambda| \hat{D}_t(\lambda)$$

$$= \left( \sum_{y^{(i)} \in \lambda} P_{i,t} \frac{r^{(i)} - r_{mean}(\mathbb{G})}{r_{std}(\mathbb{G})} \right) - \sum_{y^{(i)} \in \lambda} D_{i,t}$$

$$= \sum_{y^{(i)} \in \lambda} (P_{i,t} \cdot a_i) - D_{i,t}$$

$\square$

## C. Experimental Setup

All experiments were conducted on a single NVIDIA H100 GPU. We trained all models with 24 gradient accumulation steps per step and a generation batch size of 6. The models were evaluated on the validation split every 25 training steps.

We additionally hard-coded the generation procedure to halt after "\boxed{...}" was detected: this was to prevent the model from generating multiple boxed answers for a single prompt.

## D. Table 3

| Model | $\beta$ | Version | AIME24 | MATH-500 | AMC23 | Minerva | OB | Avg. |
|---|---|---|---|---|---|---|---|---|
| Qwen | 0.0 | Base | +0.1667 | +0.0160 | 0.0000 | -0.0074 | +0.0252 | +0.0401 |
| | | GRPO | +0.0334 | +0.0800 | +0.1250 | +0.0404 | +0.0904 | +0.0738 |
| | 0.04 | Base | +0.2000 | +0.0040 | +0.0500 | 0.0000 | +0.0282 | +0.0564 |
| | | GRPO | +0.1000 | -0.0320 | +0.0500 | +0.0368 | +0.0178 | +0.0345 |
| Llama | 0.0 | Base | 0.0000 | +0.0340 | +0.0500 | +0.0037 | +0.0059 | +0.0187 |
| | | GRPO | 0.0000 | +0.0320 | +0.0500 | -0.0036 | +0.0015 | +0.0160 |
| | 0.04 | Base | +0.0333 | +0.0280 | 0.0000 | +0.0257 | -0.0074 | +0.0159 |
| | | GRPO | +0.0333 | +0.0380 | -0.1000 | +0.0220 | -0.0044 | -0.0022 |

*Table 3.* Difference in accuracy between the $\lambda$-GRPO and GRPO-trained models, and their corresponding base models. Positive differences (i.e. $\lambda$-GRPO outperforms the comparison model) are highlighted in green; negative differences (i.e. the comparison model outperforms $\lambda$-GRPO) are highlighted in red. For example, the top-most entry in the AIME24 column indicates that the $\lambda$-GRPO Qwen model with $\beta = 0.0$ outperformed the base DeepSeek-R1-Distill-Qwen-1.5B by 0.1667 on the AIME24 benchmark.

## E. $\mathcal{B}(\mathbb{G})$ Structure Examples

The following (Figures 7, 8, 9) represent $\mathcal{B}(\mathbb{G})$ structures on groups generated during the group size 6 trial of the experiment in Section 3.2. A full trajectory is reconstructed by tracing the unique path from the root to a terminal node. The root (red) corresponds to the prompt/query $x$. Terminal nodes (yellow) denote singleton process steps $\{y^{(i)}\}$; each non-terminal node $\lambda$ (white; including the root) denotes the process step corresponding to the set of all terminal nodes dominated by $\lambda$. For the sake of presentation, overlong terminal steps are truncated with "...".

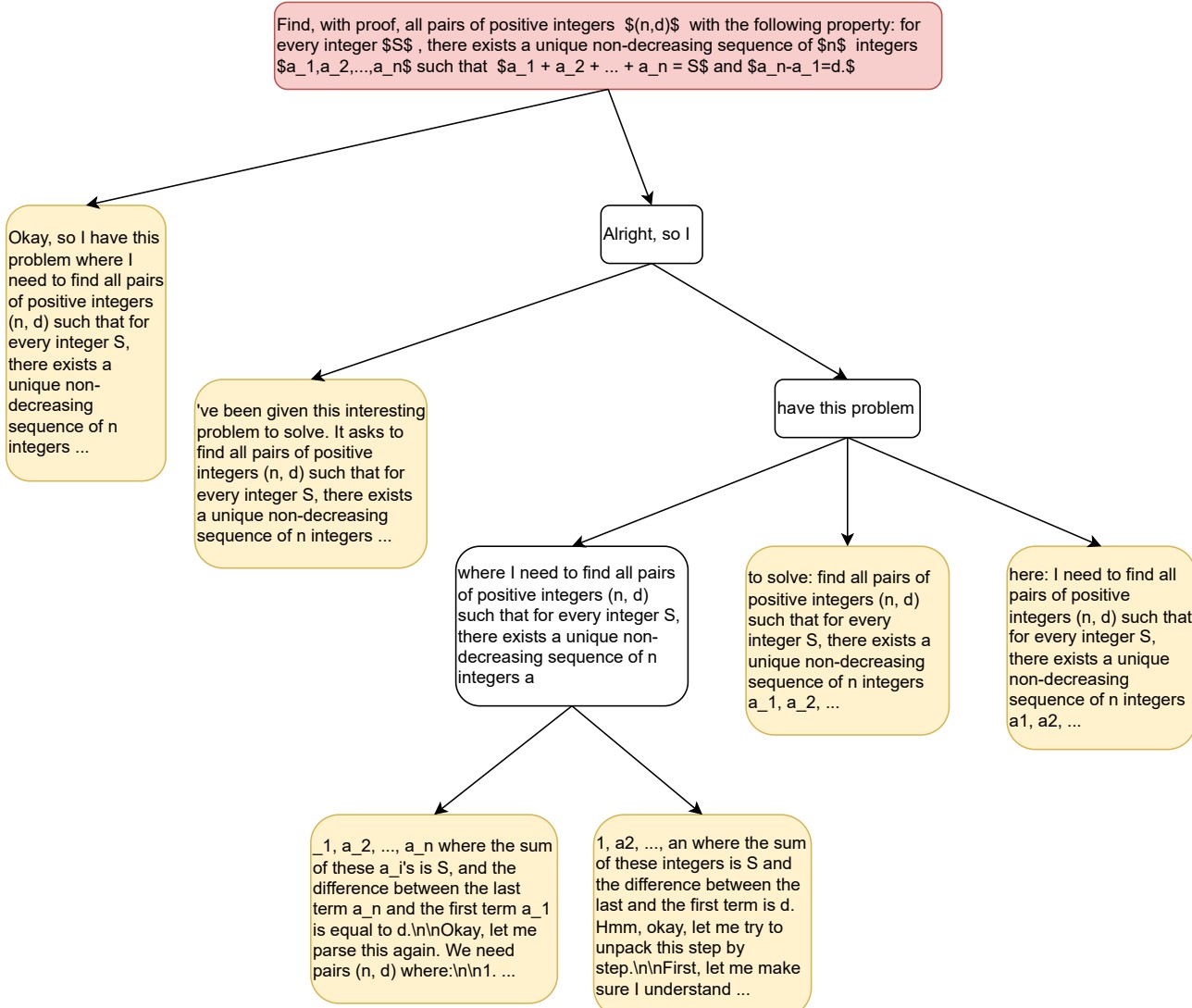

*Figure 7.* $\mathcal{B}(\mathbb{G})$ structure from step 1 (see the beginning of Appendix E for additional details).

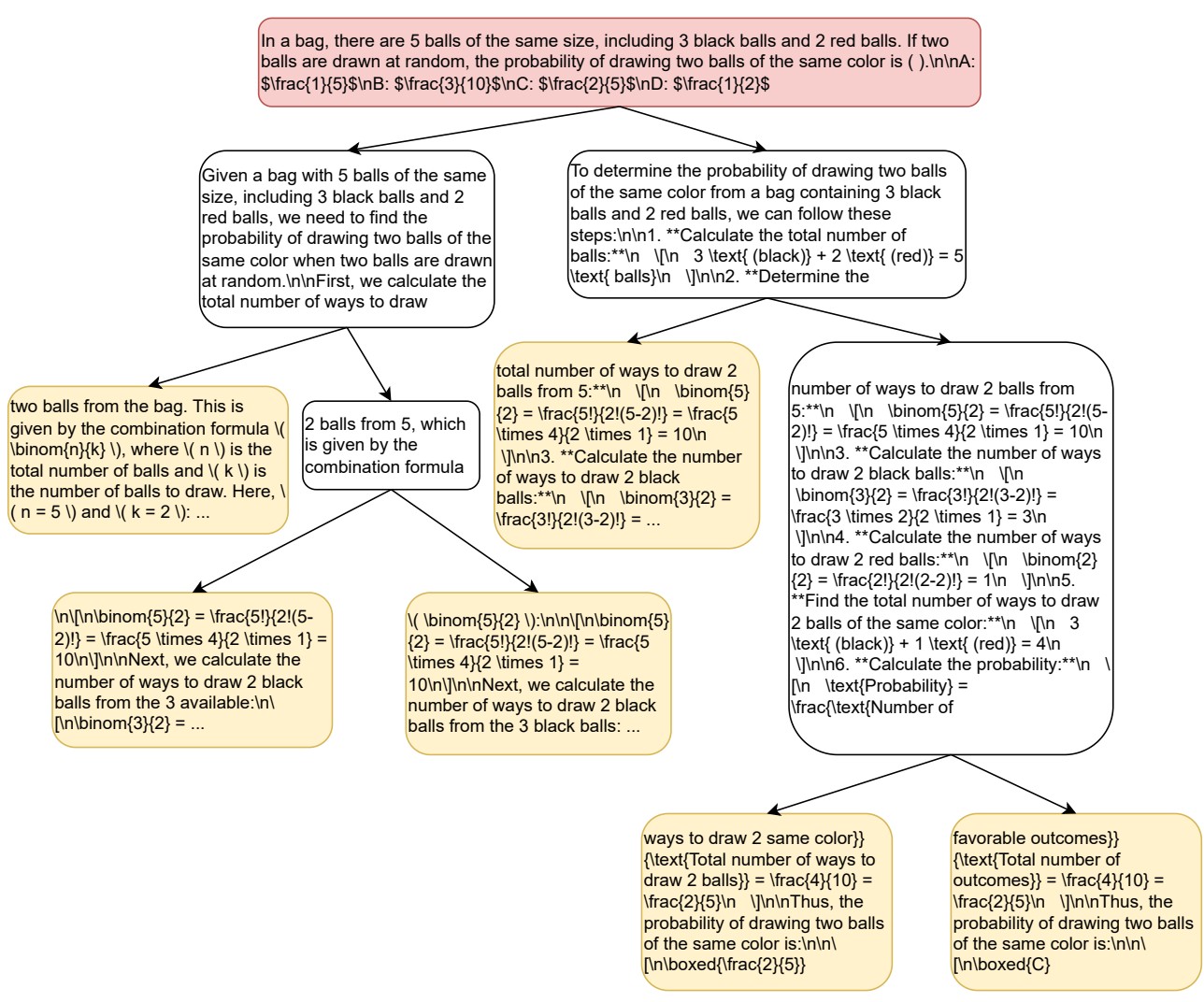

*Figure 8.* $\mathcal{B}(\mathbb{G})$ structure from step 1001 (see the beginning of Appendix E for additional details).

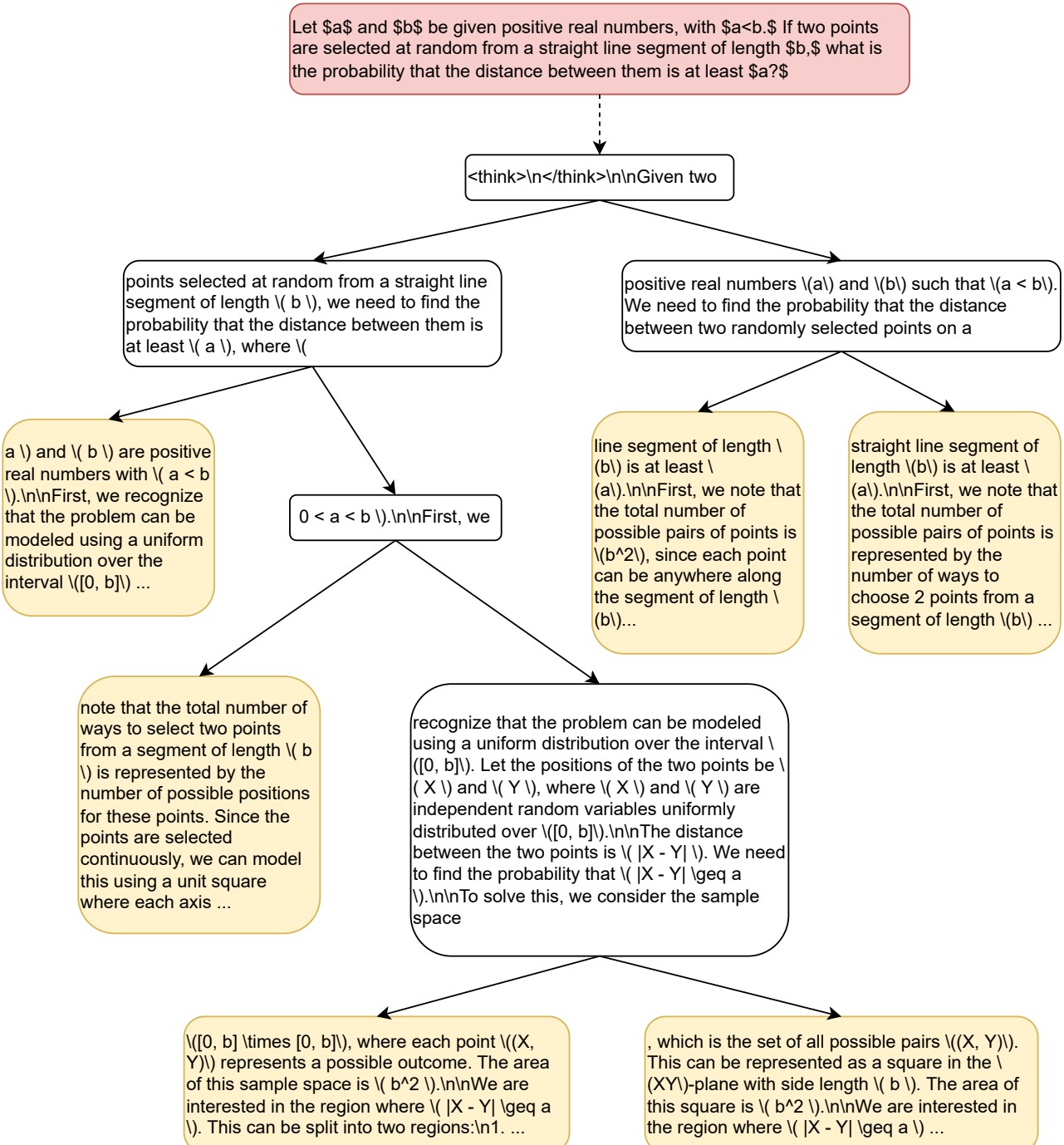

*Figure 9.* $\mathcal{B}(\mathbb{G})$ structure from step 1673 (see the beginning of Appendix E for additional details).

