# OpenReview forum: "GRPO is Secretly a Process Reward Model"
_ICML.cc/2026/Conference — ICML 2026 regular_

### Official Review · Reviewer_qYoM · 2026-03-07

**Soundness:** 3
**Presentation:** 2
**Significance:** 2
**Originality:** 3
**Overall Recommendation:** 4
**Confidence:** 4

**Summary:**

The paper reformulates GRPO algorithm as an RL approach with process reward, and identifies that GRPO objective assign imbalance weights to process steps. This motivates the $\lambda$-GRPO algorithm, which scales the advantage of each process group by the size of the group, so that each process group is assigned uniform weight. The paper shows empirically that prefix sharing is common in math reasoning tasks, and tests the algorithm against GRPO on 2 models and multiple reasoning benchmarks.

**Compliance With Llm Reviewing Policy:**

Affirmed.

**Final Justification:**

I would still maintain my score for the theory part of the paper. Nevertheless, the rebuttal from the authors suggest that experiments should be conducted under a more rigorous setup.

**Key Questions For Authors:**

See weakness above.

**Limitations:**

Yes

**Strengths And Weaknesses:**

**Strengths**
1. The paper makes a solid theoretical claim that GRPO is equivalently to an RL algorithm with process reward. This is sufficient to motivate the $\lambda$-GRPO algorithm.
2. The paper views GRPO from the novel perspective of process reward, which is inspiring for other RLVR algorithms as well.
3. The paper illustrates the process reward framework through a concrete example, which enhances readability.

**Weakness**
1. The improvement of $\lambda$-GRPO over GRPO is insignificant according to Table 1. This casts doubt on the practical effect of $\lambda$-GRPO.
2. The curves in Figure 4 is highly unstable. It does not make sense to compare peak accuracy as it can be the result of noise. The final performance of GRPO and $\lambda$-GRPO are similar in all 4 subfigures.
3. The paper restricts the theory and experiments to the GRPO algorithm. However, the proposed method should work on other RLVR algorithms as well, e.g., RLOO, GSPO, DAPO. The significance of the paper could be improved if $\lambda$ scaling with other algorithms were empirically tested.
4. The paper allocates too much space in introducing notations for the process reward framework. While it makes the proof rigorous, it hinders the reader from understanding the core of the proof, and most importantly, takes up room for putting empirical results. Since the process reward framework is actual a natural conversion from outcome reward to process reward, I personally recommend introducing the  framework in an informal way and put the rigorous proof in the appendix.
5. Section 4.1 is an important section that points out the imbalanced weight assignment in GRPO. However, the complex mathematical equations fail to convey the key idea. Personally, I suggest adding an informal proposition, claiming that the weight of each process group is $O(|\lambda|)$ (if I understand correctly). The illustration with the example of Figure 2 is perfect. It can be even better if $\lambda$-GRPO is compared with GRPO under this example, so that is more clear to understand the difference.
6. I would like to see an experiment, in which the confidence of the prefix from a correct response decreases under GRPO, while such error does not happen with $\lambda$-GRPO. This is a crucial empirical evidence that $\lambda$-GRPO solves the issue of GRPO. The task can be a simple, synthetic task.
7. Minor issue: The notation $\Sigma^*$ in Definition 1 should be explained. Empirical researchers might not be familiar with such notations.

Overall, the paper provides a novel viewpoint that connects outcome reward based RL algorithms with the process reward based counterparts, and conducts a rigorous theoretical proof. The paper can be further improved by increasing readability of the analysis and adding deeper empirical tests.

---

> ### Author Rebuttal · Authors · 2026-03-30
>
> Thank you for your review.
>
> 1. We would like to point out that Qwen with $\lambda$-GRPO improves over standard GRPO in Table 1 by +7.39 (+15%) mean performance with $\beta=0.0$ and +3.45 (+6%) with $\beta=0.04$. Llama with $\lambda$-GRPO and $\beta=0.0$ improves over standard GRPO by +1.59 (+19%). Although Llama with $\lambda$-GRPO scores slightly worse than standard GRPO for $\beta=0.04$ (0.0973 vs 0.0996), both of these scores are lower than those of the $\lambda$-GRPO Llama model with $\beta=0.0$ (0.1001).
>
> 2. We agree that these curves are unstable. However, $\lambda$-GRPO consistently reaches peak validation performance first, across four different runs. This indicates that $\lambda$-GRPO can be leveraged to yield better performance in the case of a limited compute budget.
>
> 3. We agree that this is an interesting direction for future work, but felt that an investigation of this was outside the scope of this paper.
>
> 4-5. We can certainly adjust the presentation to incorporate this feedback in the camera-ready version.
>
> 6. See our response to Reviewer c18X (the section titled "Anti-Exploitation Experiment").

---

> > ### Author Rebuttal · Reviewer_qYoM · 2026-04-01
> >
> > Follow up questions:
> > - (Weakness 1.) For each entry in Table 1, do you choose the highest validation accuracy during the training process, or the accuracy at a fixed step (e.g., end of the epoch)?
> > - (Weakness 2.) Reaching the peak earlier does not imply the algorithm is better. Given instability of the curve, the peak can be merely a result of randomness. I suggest repeating the experiment under multiple seeds and averaging the curves. By observing the convergence performance of Figure 4, my conclusion is that $\lambda$-GRPO performs similarly as GRPO.
> >
> > Nevertheless, the main contribution of the paper lies in its theoretical analysis and the novel perspective of connecting GRPO with PRM. Given the fact that most RL algorithms for reasoning have similar performance, I would still keep my score.

---

> > > ### Author Response · Authors · 2026-04-02
> > >
> > > 1. We evaluate on the checkpoint with the maximum validation accuracy. This validation set is drawn from withheld training data, not the data we evaluate on in Table 1 (see L370-372, right).
> > >
> > > 2. We will conduct these experiments by the camera-ready deadline. Given our compute constraints, we are unfortunately unlikely to have results by the end of the discussion period.

---

### Official Review · Reviewer_p5wh · 2026-03-12

**Soundness:** 3
**Presentation:** 3
**Significance:** 3
**Originality:** 3
**Overall Recommendation:** 4
**Confidence:** 3

**Summary:**

The paper proposes to reinterpret GRPO as a non-trivial PRM. The core contributions are:
- establishing a link between PRMs and GRPO, specifically arguing that the implicit model is not an ORM and the prefix structures are varied in real-world condition.
- A formulation of a RL aware PRM objetive.
- A tweak on the token level loss of GRPO.
The proposed approach, \lambda-GRPO, is tested against baseline GRPO through the fine tuning of Qwen and LLama models. The proposed method beats out vanilla GRPO in most cases for well recognized benchmarks (AIME24, MATH-500, AMC23, Minerva).

**Compliance With Llm Reviewing Policy:**

Affirmed.

**Key Questions For Authors:**

1. Did you experiment with different group size in your Experiments setup (section 5)?
2. Following up on this but given the results in Appendix E it would be interesting to see the influence of group size on the proportion of non-trivial trajectories.

**Limitations:**

Yes

**Strengths And Weaknesses:**

Soudness: The justification of the interpretation of GRPO as PRM (or rather, that the algorithm incorporates a PRM as is) is well substantiated. Theorem 1 in particular is the theoretical hinge on which the rest of the paper rests. Section 3.2 delivers an empirical analysis to further strengthen the analogy. The reasoning behind cancelling out the lambda term is also clearly laid out.

Presentation: The paper is well written, states the intended demonstration and proposes an approach in line with the previous conclusions. Figure 2 really helped getting through the notations, are some of the sections are quite dense (3.1 for instance).

Significance: Splitting the contributions in two between theoretical and empirical, the paper proposes elegant proofs to concepts that can be reemployed in future algorithms. Specifically, the link between PRMs and GRPO deepens the understanding of real-world GRPO training and the incurred step-level reward structure, which is not an intuitive results given how the trajectory reward is calculated (hence "hidden"). The applied aspect of the paper is the weaker leg: one of the claim of the abstract is higher validation accuracy, and Figure 4 shows a very incremental (and not sustained) peak. Although the change to GRPO's loss is well justified, and shows to be effective, it is a tweak. The results of table 1 are also only discussed as a whole. Degradation of performance on OB with LLam and the large gain in performance of Qwen on AIME24 would be good candidate for practical analysis.

Originality: The idea reminds me of the different papers on dense GRPO trying to introduce intermediary rewards. I think the approach of enabling an already existing step-level structure is quite original and could be combined to the aforementioned methods for more tractable experimental results.

---

> ### Author Rebuttal · Authors · 2026-03-30
>
> Thank you for your review.
>
> 1. We did not conduct experiments with different group sizes due to our limited compute budget: we selected a group size of 6 for the experiments in Section 5 because a group size of 6 outperformed a group size of 36 (in terms of maximum validation reward) with the same compute budget for standard GRPO in our experiments in Section 3.2.
>
> 2. Our experiments in Section 3.2 touch on this, at least for standard GRPO: both group sizes had effectively zero trivial structures (see L268-274, right).

---

> > ### Author Rebuttal · Reviewer_p5wh · 2026-04-02
> >
> > Fair. I do not need to update my score.

---

### Official Review · Reviewer_qtLV · 2026-03-13

**Soundness:** 4
**Presentation:** 4
**Significance:** 4
**Originality:** 4
**Overall Recommendation:** 5
**Confidence:** 3

**Summary:**

The paper theoretically proves the relationship between GRPO and PRM, and proposes λ-GRPO to address some defects of GRPO, which is validated on several downstream reasoning tasks.

**Compliance With Llm Reviewing Policy:**

Affirmed.

**Final Justification:**

I still do not find the source code of this work.

**Key Questions For Authors:**

The appendix mentions resource limitations for training model sizes, and hopes to see the algorithm's application validated on a wider range of models in the future.
Additionally, I have not found the location of the open-source code and models, and hope that they can be provided in the paper to facilitate more people in jointly expanding the application scale of the algorithm and validating it on different datasets.

**Limitations:**

yes

**Strengths And Weaknesses:**

Strengths:
1、It reveals a novel perspective on a common algorithm. Although it does not propose a completely new algorithm, this reinterpretation should be considered innovative.
2、It provides theoretical proofs and experimental evidence for the new theoretical interpretation under different conditions.
3、The improvement idea behind λ-GRPO is simple yet effective, and its approach holds valuable lessons for migration to other scenarios.
Weaknesses:
The appendix mentions resource limitations for training model sizes, and hopes to see the algorithm's application validated on a wider range of models in the future.
Additionally, I have not found the location of the open-source code and models, and hope that they can be provided in the paper to facilitate more people in jointly expanding the application scale of the algorithm and validating it on different datasets.

---

### Official Review · Reviewer_c18X · 2026-03-13

**Soundness:** 3
**Presentation:** 3
**Significance:** 2
**Originality:** 3
**Overall Recommendation:** 4
**Confidence:** 3

**Summary:**

This paper shows that, under common training choices, the GRPO objective with outcome rewards is mathematically equivalent to optimizing a PRM-aware objective equipped with a Monte-Carlo PRM that assigns step-level rewards to shared-prefix subsequences within each prompt group. Building on this perspective, the authors identify a frequency-bias in GRPO’s advantage aggregation over shared prefixes and propose a simple fix, λ-GRPO, that normalizes token-level updates by the number of trajectories sharing a process step. Empirically, they (i) demonstrate that non-trivial shared-prefix structures arise frequently during GRPO training and (ii) show consistent gains of λ-GRPO over standard GRPO on several math reasoning benchmarks with small open models.

**Compliance With Llm Reviewing Policy:**

Affirmed.

**Key Questions For Authors:**

1. How does the equivalence and the λ-GRPO normalization behave when μ > 1 (multiple epochs/updates per batch) and clipping is active? Can you provide theoretical or empirical evidence in these regimes?
2. What is the computational overhead of building B(G) and computing |λ| per token at training time? Please provide wall-clock measurements and scaling behavior with group size, sequence length, and batch size.
3. Does λ-GRPO correspond to optimizing an explicit objective (e.g., a PRM objective with uniform weighting over process sets)? If not, can you characterize bias/variance tradeoffs introduced by 1/|λ| weighting?
4. How sensitive are results to group size and sampling temperature? Your structure analysis considered group size 36; could you report λ-GRPO performance for different group sizes and temperatures?
5. Can you provide multi-seed results and standard deviations for the main tables? Some confidence intervals look like per-benchmark binomial CIs rather than across training seeds.
6. How does λ-GRPO interact with different KL coefficients and reference models, and does it remain beneficial when KL is strong? Any evidence on larger models (e.g., ≥7B) or other datasets?
7. Could you compare to or discuss methods like FlexRec that combine step-level credit assignment with uncertainty-aware scaling? Would adding uncertainty weighting on top of λ-GRPO yield further gains?
8. In the toy counterexample demonstrating anti-exploitation, can you design a controlled synthetic environment to isolate and measure this effect empirically?

**Limitations:**

- The main theoretical equivalence relies on a restricted setting, especially the assumption of a single update per batch ($\mu = 1$) and effectively inactive clipping. Since many practical GRPO implementations use multiple updates per batch and nontrivial clipping, it remains unclear how well the equivalence result and the proposed normalization extend to more standard training regimes.

- The argument for the frequency bias in GRPO is intuitive and supported by toy examples, but it is still mostly illustrative. A more formal analysis of when this bias systematically harms learning, or a controlled synthetic experiment isolating the effect, would strengthen the claim.

- The paper does not establish whether $\lambda$-GRPO is an unbiased estimator of any explicit objective. Dividing updates by $|\lambda|$ is intuitively appealing, but the resulting bias-variance tradeoff and the exact optimization target are not analyzed.

- The experimental evaluation is somewhat limited in scope. Results are restricted to relatively small models and a single RL training dataset, so it is not yet clear whether the observed gains will hold at larger scales or across more diverse tasks.

- Robustness reporting is limited. The paper does not clearly provide multi-seed training statistics, and the reported confidence intervals do not appear to fully address the high variance typically seen in RL-style reasoning experiments.

**Strengths And Weaknesses:**

**Strengths**
- Formalizes a clear and intuitive equivalence between GRPO with outcome-level rewards and a PRM-aware optimization objective using a Monte-Carlo PRM over shared prefixes.
- Identifies a non-obvious frequency-induced bias in GRPO’s token-level aggregation when many trajectories share the same step, and proposes a simple, principled normalization by the step frequency.
- Demonstrates improvements of λ-GRPO over GRPO across multiple downstream reasoning benchmarks and two model families, with faster attainment of peak validation accuracy.
- The proof sketch and supporting lemma are easy to follow conceptually; the phenomenon is well-motivated through concrete counterexamples.
- The λ-GRPO modification is simple to implement, appears to yield better sample efficiency, and could be impactful for large-scale training where PRMs are costly.

**Weaknesses**

- The equivalence proof relies on mild but constraining assumptions: DAPO-style token-level loss and μ = 1 update per batch (so that ratio clipping is irrelevant). Many practical GRPO recipes use multiple epochs/updates per batch and nontrivial clipping; it is unclear how (or whether) the equivalence and the proposed fix extend to μ > 1 and active clipping.
- The theoretical discussion of “anti-exploration” and “anti-exploitation” effects is primarily illustrative via a toy example; a more formal characterization (or synthetic controlled experiments) would strengthen the argument.
- Limited sensitivity analyses: no evaluation with larger group sizes for λ-GRPO, different temperatures, or different μ and clipping settings. Section 3.2 explores group size effects for structure diagnostics but not for λ-GRPO’s performance.
- Checkpoint selection uses a held-out split from the training source distribution (OpenRS); while common, this selection criterion may favor methods that overfit validation-like samples.
- Minor indexing/notation typos (e.g., t vs t+1 equality conditions) and occasional reliance on figures that are hard to parse from text could be improved.
- Connections to recent GRPO variants that introduce step-level or item-level credit assignment and uncertainty weighting (e.g., FlexRec’s item-level swap rewards with GRPO and uncertainty-weighted advantages) are not discussed in depth. Such works speak to similar design space (token/step-level credit, group normalization) and could contextualize λ-GRPO’s benefits and tradeoffs.

---

> ### Author Rebuttal · Authors · 2026-03-30
>
> Thank you for your review.
>
> 1. It is possible to derive an analogous PRM and PRM-aware RL objective $\hat{L}$ for $\mu>1$, such that the *gradient* of $\hat{L}$ is equivalent to that of $L_{GRPO}$: i.e. $\nabla_\theta \hat{L}(G) = \nabla_\theta L_{GRPO}(G)$ for all groups $G$. The derivation of $\hat{L}$ is a lot less clean than the case $\mu=1$ (and we only have an equivalence of gradients, not an exact equivalence), so we left an analysis of this to future work. We can include the derivation of $\hat{L}$ in the appendix of the camera-ready version if the reviewers believe it would improve the overall paper.
>
> 2. The time complexity of building $B(G)$ and computing the per-token lambda-scaling terms is $O(k^2 n)$, where $k$ is the group size and $n$ is the maximum sequence length. However, the multiplicative constant hidden in the big-$O$ notation here is very small. We measured wall-clock time on the trajectories from our experiments in Figure 3: for group sizes $k = 6$ and $k = 36$, the added latency is 1.19e-7 and 1.21e-7 seconds/token, respectively (on a single Intel i7 CPU). This amounted to 8.38 and 10.27 total seconds of latency (respectively) for each >1-week-long training run.
>
> 3-6. We will attempt to conduct experiments to address these questions by the camera-ready deadline.
>
> 7. FlexRec seems to be fairly orthogonal to our approach, in that it explicitly computes and assigns subtrajectory-level rewards with a modified GRPO objective, along the lines of Feng et al. (2025) (see L397-399, right). We can certainly include a discussion of this work in the camera-ready version: it was not included in this submission because the FlexRec paper was released on ArXiv six weeks after the ICML deadline.
>
>
>
> **Anti-Exploitation Experiment:**
>
> We created a toy, synthetic task to more rigorously evaluate our anti-exploitation claims in Section 4.2. We constructed a depth-4 binary tree as the environment, and trained GPT-2-small to traverse the tree from the root to a leaf node, with two possible actions: "L" (left) and "R" (right)---all other tokens were masked.
>
> One path $T$ was sampled as the target (e.g. $T$ = "LLRL"), and given the highest reward (1.0). The first $n$ steps of $T$ served as the shared prefix, and all paths beginning with $T$[:$n$] (aside from $T$ itself) were given the lowest reward (-1.0). All other paths in the tree were given a reward of 0.7. See here for an example with $n=2$: https://freeimage.host/i/tree1.BJZs3Sn
>
> This setup creates a scenario similar to our anti-exploitation example in Section 4.2: if trajectories $T$="LLRL", $X$="LLLR", and $Y$="LLRR" are all in the generated group, then the mean reward of {$T, X, Y$} is negative, and therefore the process reward corresponding to the sub-trajectory "LL" is negative.
>
> If our hypothesis is correct, then GRPO should fail to converge on the maximum-reward path $T$, and instead learn to generate the safer paths with 0.7 reward, while $\lambda$-GRPO converges on $T$.
>
> We evaluated this setup across a range of hyperparameter configurations, conducting a grid search across $n\in$ {1,2}, *group_size* $\in$ {8,16}, and $lr\in$ {1e-5,5e-5}, each for five seeds with 250 steps. We found similar results in all cases, and plotted the results across all five seeds for $n=1$, *group_size* $=16$, $lr=$ 1e-5 here: https://freeimage.host/i/reward1.BJZbTlV
>
> As you can see, standard GRPO converges on the safer 0.7 reward paths, while 3/5 $\lambda$-GRPO runs converge on the maximum-reward path. Viewing the pre-group frequency of target path generation (https://freeimage.host/i/target-frequency1.BJt9GQn), we see that GRPO-trained models do in fact generate the target path during early training: they did not fail to discover this path---they failed to exploit it. Only one out of the five $\lambda$-GRPO models entirely fails to exploit the target path.
>
> (we will of course polish the linked figures for the camera-ready version)

---

> > ### Author Rebuttal · Reviewer_c18X · 2026-04-04
> >
> > Thank you for the rebuttal. Most of my concerns have been resolved, and I appreciate the authors’ thoughtful responses and effort in addressing them. I encourage the authors to incorporate these clarifications into the revised version.

---

### Decision · Program_Chairs · 2026-04-30

**Decision:**

Accept (regular)

**Comment:**

This paper presents a novel and useful theoretical perspective on GRPO, showing that under common training assumptions it can be interpreted as optimizing a PRM-aware objective with an implicit Monte Carlo process reward model. Reviewers generally found this reinterpretation insightful and technically interesting, as it connects outcome-level rewards in GRPO to a hidden step-level reward structure and motivates a simple modification, λ-GRPO, to correct an imbalance in token-level credit assignment.

Multiple reviewers found the equivalence argument convincing and the proposed normalization principled and easy to implement. The empirical section also provides encouraging evidence that shared-prefix process structure is common in practice. The rebuttal further strengthened the paper by clarifying computational overhead, and adding a synthetic experiment supporting the claimed anti-exploitation effect.

At the same time, the paper has clear limitations. The cleanest theoretical result holds in a restricted setting, especially the single-update-per-batch regime with inactive clipping, and it remains unclear how broadly the equivalence extends to more standard practical GRPO settings. On the empirical side, the gains are not uniformly large, the training curves are unstable, and reviewers reasonably asked for stronger multi-seed analysis and broader evaluation across scales and training regimes.